# Unsupervised Learning-Based Non-Invasive Fetal ECG Muti-Level Signal Quality Assessment

**DOI:** 10.3390/bioengineering10010066

**Published:** 2023-01-04

**Authors:** Xintong Shi, Kohei Yamamoto, Tomoaki Ohtsuki, Yutaka Matsui, Kazunari Owada

**Affiliations:** 1Graduate School of Science and Technology, Keio University, Yokohama 223-8522, Japan; 2Department of Information and Computer Science, Keio University, Yokohama 223-8522, Japan; 3Atom Medical Co., Tokyo 113-0021, Japan

**Keywords:** non-invasive fetal electrocardiogram, multi-level classification, signal quality assessment, unsupervised learning, autoencoder

## Abstract

Objective: To monitor fetal health and growth, fetal heart rate is a critical indicator. The non-invasive fetal electrocardiogram is a widely employed measurement for fetal heart rate estimation, which is extracted from the electrodes placed on the surface of the maternal abdomen. The qualities of the fetal ECG recordings, however, are frequently affected by the noises from various interference sources. In general, the fetal heart rate estimates are unreliable when low-quality fetal ECG signals are used for fetal heart rate estimation, which makes accurate fetal heart rate estimation a challenging task. So, the signal quality assessment for the fetal ECG records is an essential step before fetal heart rate estimation. In other words, some low-quality fetal ECG signal segments are supposed to be detected and removed by utilizing signal quality assessment, so as to improve the accuracy of fetal heart rate estimation. A few supervised learning-based fetal ECG signal quality assessment approaches have been introduced and shown to accurately classify high- and low-quality fetal ECG signal segments, but large fetal ECG datasets with quality annotation are required in these methods. Yet, the labeled fetal ECG datasets are limited. Proposed methods: An unsupervised learning-based multi-level fetal ECG signal quality assessment approach is proposed in this paper for identifying three levels of fetal ECG signal quality. We extracted some features associated with signal quality, including entropy-based features, statistical features, and ECG signal quality indices. Additionally, an autoencoder-based feature is calculated, which is related to the reconstruction error of the spectrograms generated from fetal ECG signal segments. The high-, medium-, and low-quality fetal ECG signal segments are classified by inputting these features into a self-organizing map. Main results: The experimental results showed that our proposal achieved a weighted average F1-score of 90% in three-level fetal ECG signal quality classification. Moreover, with the acceptable removal of detected low-quality signal segments, the errors of fetal heart rate estimation were reduced to a certain extent.

## 1. Introduction

For evaluating the health of fetuses, non-invasive fetal electrocardiograms (FECGs) made from abdomen surface electrodes are commonly used. Estimating the fetal heart rate (FHR) is essential for prenatal fetal monitoring. The quality of the FECG signals has a significant impact on how accurately the FHR is estimated using FECG data. In other words, as the quality of FECG degrades associated with noise interferences such as maternal ECG (MECG) and fetal movements, reliably calculating FHR becomes challenging [1]. By removing or interpolating the FHRs predicted from low-quality recordings, signal quality assessment (SQA) can help increase the accuracy of FHR estimation.

Since there are not many suitable resources, tools, or skills accessible for medical data analysis, many machine learning and deep learning techniques are developed for several medical research topics recently [2], including heart rate estimation [3], disease diagnosis [4], blood pressure estimation [5], patient activity recognition [6], biomedical signal quality assessment [7], etc. The two primary task categories in machine learning are supervised and unsupervised learning. The primary distinction between the two types is that supervised learning is carried out with the aid of ground truth. As a result, the objective of supervised learning is to discover a function that, given a sample of data and the intended outputs, most closely approximates the connection between input and output shown in the data. Contrarily, the objective of unsupervised learning is to deduce the inherent structure existing within a set of data points because it lacks labeled outputs. Although various adult ECG SQA approaches based on both supervised and unsupervised learning have been put forth [7,8,9,10], little study based on supervised learning has been done on FECG SQA [11,12,13]. The existing FECG SQA methods are all based on supervised machine learning, which requires large datasets with annotations. Moreover, many features used in existing FECG SQA methods highly depend on the results of MECG cancellation and FHR estimation.

In this proposal, we introduce the FECG SQA approach based on unsupervised learning to increase the accuracy of FHR estimation. Based on earlier research [9,10], an AE-based feature for assessing the FECG signal quality is introduced. To reconstruct 2D spectrograms made from the processed FECG signals, we train a fully convolutional network (FCN)-based AE, and from the AE-based reconstruction error, an AE-based feature is derived. To determine the FECG signal quality, entropy-based characteristics [14,15,16,17], statistical features [18,19], and frequently utilized ECG SQIs [7] are also extracted. The importance of each extracted feature is evaluated based on the multi-class feature selection (MCFS) method [20]. We select several top significant features, and these selected features are used to differentiate between segments of the high-, medium-, and low-quality by feeding them into a binary classifier based on the self-organizing map (SOM). A preliminary version of this work has been reported [21].

We assessed the performance of our proposal using the actual FECG datasets extracted from 10 experimental subjects. As a result, a 90% average F1-score in identifying high-, medium-, and low-quality FECG signal segments were shown for our proposed FECG SQA. In addition, comparing the circumstances with and without the removal of detected low-quality signal segments, it can be proved that with the acceptable removal of detected low-quality signal segments, the errors of FHR estimation were reduced to a certain extent. The main contributions of our work are as follows:

(i) A novel AE-based feature for SQA is introduced. It can be proved that the novel feature plays a significant role in the FECG SQA task.

(ii) To the best of our knowledge, this is the first work using an unsupervised learning-based FECG SQA method to identify high-, medium-, and low-quality FECG signals, which does not require large datasets with annotations.

The following is the outline of this paper. Section 2 summarizes the existing adult and fetal ECG SQA methods in the literature. Some preliminaries for this research are explained in Section 3. The proposed method is described in Section 4. The experiment results and limitations of our research are discussed in Section 5. The paper is concluded in Section 6.

## 2. Related Work

With regard to adult ECG SQA, a support vector machine (SVM)-based low-quality ECG signal identification method has been proposed by Clifford et al. [7]. This approach made advantages of signal quality indicators (SQIs), such as spectral energy distribution, higher-order moments, inter-channel, and inter-algorithm agreement. Although the accuracy of this method is very high, the work can only classify two quality levels. In practical applications, multi-level classification of signal quality can be applied in different scenarios, such as selecting to remove the signal data of one or more quality levels according to different requirements of quality standards. Another method for ECG SQA [8] uses SVM to categorize ECG signal quality into five categories based on thirteen SQIs. These two approaches [7,8] introduced several effective SQIs for ECG SQA, however, the simple machine learning classifier SVM limited the improvement of classification accuracy. Recently, some deep learning-based ECG SQA methods have been proposed to further improve the accuracy of ECG SQA. An ECG SQA, which is based on a convolutional neural network (CNN) with two quality labels (high- and low-quality) as outputs and 2D images created using the continuous wavelet transform (CWT) as inputs, has been introduced by Huerta et al. [9]. Though the accuracy of ECG SQA has been improved compared with [7,8], this is still a supervised learning-based method requiring data annotations which needs the experts and a commitment of time and energy. To achieve unsupervised learning-based ECG SQA for solving the problem of the limited labeled datasets, two autoencoder (AE)-based SQIs related to reconstruction error and confidence have been provided by the approach [10] for evaluating ECG quality. However, the authors only compared the novel AE-based SQIs with other benchmark-SQIs, and no classifier was used in this work. Moreover, only the time domain was considered in this work; the frequency domain was not analyzed. In terms of FECG SQA, containing more noise effects than the adult one, a traditional technique [11] chose the best FECG recording for FHR estimation using a ten-feature decision tree (DT). This work used FHR as one of the features for evaluating the FECG signal quality, which highly depends on the algorithms and results of FHR estimation. This feature may not be trustworthy if the results of the FHR estimation are unreliable. Furthermore, this paper only analyzed long-time duration recordings rather than signal segments, so it is difficult to use this method for removing some noisy signal segments with short-time duration. Andreotti et al. [12] have introduced several innovative SQIs to evaluate how effectively the MECG is suppressed for classifying five FECG quality levels. As well as [11], the effectiveness of the novel features introduced in this paper heavily depends on the reliability of the MECG cancellation algorithm. Fotiadou et al. [13] established a CNN-based classifier to determine if an FHR estimate is trustworthy or not to exclude the erroneous FHRs obtained from FECG signals. In this work, they set a threshold of the error of FHR estimates for labeling two reliability levels of FHR estimates related to FECG signal quality. Thus, the annotations highly depend on the results of FHR estimation. Although it has been demonstrated that existing ECG SQA algorithms can, to some extent, identify the quality of adult ECG or FECG signals, these traditional techniques still have certain drawbacks:

(i) Although the supervised machine learning-based SQA approach may achieve accurate SQA, it necessitates the use of large labeled datasets, which requires expertise and takes time.

(ii) Many frequently used SQIs [11,12] may lose their effectiveness for the FECG SQA if the results of MECG cancellation and FHR estimate are inaccurate.

Therefore, an unsupervised learning-based multi-level FECG SQA approach is proposed in our work, where the extracted features for classification do not depend on the results of MECG cancellation and FHR estimation.

## 3. Preliminaries

### 3.1. Non-Invasive Fetal ECG

During pregnancy, prenatal fetal monitoring is an essential modality to determine the growth and health of a fetus. The principle method for fetal monitoring is FHR estimation, which is generally based on fetal echocardiography, cardiotocography, or FECG. Fetal echocardiography can perform detection of atrial and ventricular activities. Nevertheless, several clinically relevant indicators, such as the QT intervals, cannot be directly accessed from fetal echocardiography. In addition, a continuous fetal echocardiographic recording requires operations by professionals and a clinical environment [22]. Cardiotocography is the recording that simultaneously contains fetal cardiac cycles and uterine contractions. It can only measure blood ejection from the ventricles and so cannot offer information on atrial activity or atrioventricular conduction. Furthermore, due to the averaging nature of the autocorrelation function employed to estimate the fetal heart rate, it is unable to extract the beat-to-beat variability of the fetal heart rate [23]. The FECG is used to assess the FHR to determine the growth and health of a fetus, thereby further detecting and preventing fetal diseases [24]. The invasive FECG measurement can obtain the signals displaying extremely clear fetal cardiac activities, which utilizes the electrodes placed on the scalp of a fetus to extract signals, as shown in Figure 1a. The invasive one can estimate FHR accurately; however, it can only be used during delivery and involves the risk of infection [25].

To address this limitation, the non-invasive FECG-based FHR estimation approach is developing for fetal surveillance. As shown in Figure 1b, the non-invasive FECG signals are recorded from the electrodes placed on the maternal abdomen using portable, wearable devices [26]. Compared with the invasive one, the non-invasive FECG performs a variety of benefits [27]: (i) low cost; (ii) remote monitoring without clinical environment and medical expertise; and (iii) long-term monitoring with high simplicity. Additionally, there are some challenges for non-invasive FECG-based FHR estimation: (i) the maternal ECG signals are contained in the FECG signals; (ii) there are various interference sources such as electrode motion artifacts and fetal movements.

### 3.2. Short-Time Fourier Transform

It is not possible to obtain temporal information of signal change when we just directly process Fourier transform to a non-stationary signal. In a period, for example, there are many components that appear and disappear at different time stamps. Only direct Fourier transform is unable to determine the order of the appearances of different components. The principle of Short-time Fourier transform (STFT) is to slice a single non-stationary signal into multiple signal segments, then take the Fourier transform of each signal segment separately, and finally combine the spectrum of each signal segment together. In a STFT signal spectrogram, the abscissa is time, the ordinate is frequency, and the color of the spectrum indicates the magnitude of frequency.

### 3.3. Autoencoder

Autoencoder(AE), as a class of neural network models, uses the unsupervised learning-based method to efficiently extract representations of high-dimensional data, and it has gained great popularity in both academic and industrial fields. The representations of the input data are extracted through the encoder in AE, and through the decoder, the input can be rebuilt [28]. The encoder and decoder blocks consist of the majority of the AE model. The neural networks in the encoder compress the input vector x into a latent representation vector z, while the neural networks of the decoder rebuild the representation vector z into the reconstructed vector $\hat{x}$. The representation vector z, which incorporates the most important components of x, such as the periodicity of fetal heart activity shown in FECG signals, may be created by the AE by compressing the input data. The representation vector z has the potential to eliminate pointless redundancy and include the most crucial information after training for a significant number of epochs. As a result, the input vector x may be rebuilt using the vector z. The reconstruction error is typically thought of as a loss function in AE, for example, the mean square error between the input and the output of the AE.

### 3.4. Self-Organizing Map

Self-organizing map (SOM) is an unsupervised learning deep neural network and is generally used for clustering and classification tasks. This technique was applied to tackle SQA challenges in previous studies [29,30]. Competitive learning is used for training, with the aim of mapping the *N*-dimensional input space into a two-dimensional output space made up of many neurons. A lattice of neurons makes up the structure of SOM. Each neuron in the grid, which contains *X* and *Y* coordinates, has a distinct topological location and a vector of weights with the same dimension as the input variables. There is no lateral connection among output neurons. Those neurons that are adjacent to one another represent clusters of neurons with related properties. As a result, through training, each neuron represents a cluster. The input training data consists of many samples with *n* dimensions. Each node in the output layer is made up of an *n* dimensional matching weight vector $W_{k}={\left(w_{1k},w_{2k},w_{3k},...,w_{nk}\right)}^{T}$, where *k* represents the *k*-th node in the output layer. A winning neuron can be found for every input sample by a trained SOM, which matches the input sample best [31]. The training process of SOM is indicated as follows:Initialize all the weight vectors randomly.Select an input sample from the training set $x={\left(x_{1},x_{2},x_{3},...,x_{n}\right)}^{T}$.By computing the Euclidean distance for each neuron *k*, compare x with the weights $W_{k}$. The neuron with the shortest distance is declared as the winning neuron.After updating the weights of neurons, the winning neuron should resemble the input vector x.The weight vectors of neighboring neurons are changed to more closely resemble the input vector. A weight vector changes more as the neuron gets closer to the winning neuron.

## 4. Proposed Method

Figure 2 displays the overall structure and workflow of our proposal. The proposed method is introduced in the following parts: (i) data acquisition, (ii) signal pre-processing, (iii) feature extraction, (iv) multi-level signal quality classification, and (v) performance evaluation.

### 4.1. Data Acquisition

The IRIS Monitor, a maternal abdominal-lead FECG machine, was used to collect data from ten healthy pregnant subjects by placing 11 electrodes on the surface of the abdomen of each subject [32]. The method proposed in [33] was utilized to extract the FECG recordings. Each individual has 11 recordings, which equates to 11 channels extracted from 11 electrodes. Each recording lasted 60 s, with a 1 kHz sampling frequency. Moreover, for the purpose of assessing the performance of our proposal, the 3 s segments were annotated by medical experts with 3 levels: high quality, medium quality, and low quality. The criterion for annotation is as follows: (i) high quality: there is almost no noise effect; (ii) medium quality: some noise can be seen in the FECG signals, however, it does not affect the heartbeat detection seriously, and the noise amplitude is far less than the heartbeat amplitude; and (iii) low quality: the noise in the signal is very obvious, and the noise amplitude is close to the heartbeat amplitude.

### 4.2. Signal Pre-Processing

Due to the polluted A/D converter output, signal pre-processing on the raw FECG data is carried out to get rid of saturated data and incorrect data [34]. Firstly, the centering and normalization operation is performed by subtracting the mean of the signals and normalizing the amplitude between [−1, 1]. Using a linear-phase Kaiser window, we apply the band pass filter between [2, 46] Hz to eliminate the noise interference. It has been confirmed by several of our experiments that this frequency range can include fetal cardiac activities. Furthermore, for spike removal, we utilize the method proposed in [35], since the outliers can interfere with the FHR estimation.

By sliding a 3 s window with 1.5 s of overlap over the FECG signals, we obtain 3 s segments of the FECG signals. Notably, this window length, 3 s, can capture temporal periodic patterns caused by fetal heartbeats because it includes many heartbeat cycles. For FHR estimation, the window length is therefore adequate to assess the FECG signal quality. In addition, the STFT, with a window length of 128 ms and a step size of 5 ms, is applied to generate a 2D spectrogram of each signal segment, and the spectrogram is extracted at a frequency of [2, 60] Hz. By selecting the most appropriate parameters for spectrogram generation, we can make sure that the spectrograms include enough information on fetal cardiac activity for subsequent signal quality analysis [36].

In Figure 3, there are several examples of the high-, medium-, and low-quality pre-processed FECG signal segments and their corresponding spectrograms. Compared with the example of the high-quality signal segment and spectrogram, the medium-quality one contains some tiny noise, which can mostly be acceptable for FHR estimation. However, as for the low-quality one, there is too much noise interference to reach an accurate FHR estimation.

### 4.3. Feature Extraction

As shown in Figure 2, an AE-based feature is generated from the segmented 2D spectrograms, and the other three categories of features are calculated from the segmented 1D processed FECG signals. All twelve features are introduced as follows.

#### 4.3.1. AE-Based Feature

Inspired by Seeuws et al. [10], we introduce a novel feature based on the spectrogram reconstruction error using AE. Seeuws et al. [10] used a fully convolutional network (FCN)-based AE for reconstructing raw ECG signal segments and the dimensions of the input and the output are both 1024 × 1. Different from their research, rather than the 1D signal segment, the 2D spectrogram is considered the input of our proposed AE. In addition, the 1D convolutional layers are replaced with 2D convolutional layers. The 2D spectrograms include more information associated with fetal cardiac activities in comparison with the 1D signal segments since they can display varieties not only in the time domain, but also in the frequency domain. It is similar to the theory mentioned in the paper [10] that the large values of reconstruction error of the spectrograms mean that the corresponding signal segments are low-quality and the smaller values mean higher quality.

As shown in Figure 4, the encoder and the decoder are almost symmetrical. A 2D convolutional (or deconvolution) layer, a batch normalization layer, and an activation layer are composed of a FCN layer. Both in the encoder and the decoder, there are 6 FCN layers. The dimension of the input and the output are both 128 × 128 × 3, where 128 × 128 indicates the size of the image and 3 indicates 3 channels for RGB coding. The convolutional layers are used for compression in the encoder and the deconvolutional layers are applied for upsampling in the decoder. All the convolutional or deconvolutional layers are performed with a 3 × 3 kernel. In order from the input to the output, the filter sizes of the convolutional and deconvolutional layers are 64, 32, 32, 32, 64, 1, 1, 64, 32, 32, 32, 64, and 1, and the stride sizes are 2, 2, 2, 2, 2, 1, 1, 2, 2, 2, 2, 2, and 1. Batch normalization, sometimes referred to as batch norm, is a technique for accelerating and improving the stability of deep neural network training by re-centering and rescaling the inputs to the layers. Batch normalization performs a transformation that keeps the output mean and standard deviation close to 0 and 1, respectively. In our proposed AE, the batch size for batch normalization layers is set as 64. Moreover, the exponential linear unit (ELU) is used for activation. We select the mean square error between the input and the output as the loss function through training in order to minimize the reconstruction error. Only the spectrograms with high quality are selected for training AE to widen the gap between high- and low-quality signal reconstruction errors. A novel AE-based feature, named AE-based mean squared error (AE_MSE), is defined as Equation (Equation 1):(1)$$AE\_MSE\left(s\right)=\frac{\sum_{l=1,m=1,n=1}^{L,M,N}{\left(s_{l,m,n}-{\hat{s}}_{l,m,n}\right)}^{2}}{L\cdot M\cdot N},$$
where L·M is the size of a spectrogram, *N* is the number of channels for RGB coding, s and $\hat{s}$ is the input vector and the output vector of AE, respectively. When the input data are of lower quality, the reconstruction error of AE increases significantly, which indicates that the signal quality decreases with increasing AE_MSE for each input 2D spectrogram.

#### 4.3.2. Entropy-Based Features

Entropy is the amount of information expected to be confirmed by observation. The amount of information confirmed is proportional to the uncertainty, so entropy can also be understood as the uncertainty contained in the system. To recognize uncertainties in biomedical signals, entropy-based features have been popular for decades. The entropy-based features are features generated from time series data using the calculation of different types of entropy. In general, low-quality FECG signals show many more irregularities and uncertainties than medium- and high-quality FECG signals. Therefore, the entropy-based features have great significance to FECG signal quality. In our research, four entropy-based features are calculated [37]:Approximate entropy (AppEn): AppEn is a common-use feature for quantifying irregularity in time series data with no knowledge about the system. Larger values of AppEn correspond to more complexity and irregularity in the data [14].Sample entropy (SampEn): To overcome the shortcomings of AppEn, including heavy dependency on the length of the recording and lack of relative consistency, SampEn was introduced. Compared with AppEn, SampEn avoids self-matches, so it can be independent of the length of recordings and extract relative consistency [15].Spectral entropy (SpecEn): Spectral entropy, based on Shannon entropy, can quantify the regularity or uncertainty of the power spectrum during a specific period. In actuality, the regularity of the power spectrum distribution is mirrored in spectral entropy. The higher SpecEn indicates a more uniform power spectrum distribution [16].Permutation entropy (PEn): A continuous time series can be transformed into a symbolic sequence using the permutation approach, and PEn is the output of the statistics of the symbolic sequences. PEn of time series data, which can be calculated simply and quickly, contains temporal information [17].

#### 4.3.3. Statistical Features

The biomedical signals can also be analyzed using statistical features. Different from the entropy-based features, the statistical features can be used to analyze the periodicity and complexity of biomedical signals based on statistical methods. The entropy-based features concentrate on the probability distributions of potential states of a system for measuring the biomedical signals, while the statistical features focus on the waveform morphology of the biomedical signals. From high- to medium- to low-quality, generally, the periodicity is decreased, and the complexity is increased. Three statistical features we utilized are explained as follows:Detrended fluctuation analysis (DFA): The main purpose of DFA is to extract long-range correlation in non-stationary time series. Many researchers have used DFA for analyzing cardiac interbeat intervals [18].Fractal dimension (FD): FD is a quantitative metric used in biomedical signal processing to gauge the complexity of discrete temporal physiological data. FD can aid in the understanding of physiological processes [19].Higuchi fractal dimension (HFD): Higuchi’s approach to FD calculation is proved to reach accurate and reliable estimation results, which is called HFD. This technique can be used to compute moving window estimates of FD for non-stationary signals by segmenting signals into brief quasi-stationary frames. It is also suited for estimating FD of segments with a short time duration of time series [19].

#### 4.3.4. ECG SQIs

Furthermore, several SQIs for adult ECG SQA [7] are used in our proposal:kSQI: The kurtosis of the ECG signal.sSQI: The skewness of the ECG signal.pSQI: The relative power in the QRS complex. pSQI is given by Equation (Equation 2):
(2)$$pSQI=\frac{\int_{5}^{15}P\left(f\right)df}{\int_{5}^{40}P\left(f\right)df},$$basSQI: The relative power in the baseline. basSQI is given by Equation (Equation 3):
(3)$$basSQI=1-\frac{\int_{0}^{1}P\left(f\right)df}{\int_{0}^{40}P\left(f\right)df},$$

### 4.4. Multi-Level Signal Quality Classification

For the sake of maximizing the accuracy of the classification task, the MCFS method [20] is used before the classification task, which is a total unsupervised method for evaluating feature importance. By means of MCFS, an MCFS score is calculated for each feature, and a feature with a larger score means that this feature plays a more significant role in the classification task. We select five important features rather than all of the extracted features to feed them into the SOM, hence the non-selected features, little associated with the FECG signal quality, cannot have a negative impact on the classification results.

Using the selected features as the input vectors, our proposal uses SOM as a three-class classifier to identify the signal segments with high, medium, and low quality. By identifying the winning neurons corresponding to the input samples, the SOM-based classifier can classify the input samples as the class of the corresponding winning neuron [38]. For the purpose of assigning a class for each neuron in the output layer, for each neuron, we randomly select three input samples whose corresponding winning neuron is this neuron and define the class of this neuron as the class represented by the majority of the three samples. Such a method only needs to annotate quality-level labels for a few signal segments, which solves the problem of the difficulty of annotation for a large dataset.

In our dataset, the FECG signal segments with medium quality are much more than the segments with high quality and low quality, which makes our dataset unbalanced. Using unbalanced data for SOM-based classification may reduce the sensitivity of detecting low-quality samples. In other words, the recall of low-quality data can be sharply decreased while low-quality data accounts for a very small part of the whole dataset. Thus, we simply balance our dataset for training by reducing the numbers of the FECG signal segments with high quality and medium quality.

Several parameter combinations for SOM are tested, and based on the quantization error of SOM [39], we selected the best parameter combinations as follows:Size of output layer: 8 × 8.Initial value of learning rate: 0.5.The number of iterations: 100,000.Neighborhood function: bubble.

### 4.5. Performance Evaluation

#### 4.5.1. For Classification Performance

Five unsupervised clustering methods are applied for the comparison of classification performance, including K-means, K-means++, hierarchy clustering, spectral clustering, and SOM. These benchmark clustering methods are selected for two reasons: (i) all these four methods are widely used for unsupervised learning-based classification tasks; and (ii) compared with some advanced deep learning-based clustering methods, the computation complexities of these four methods are at the same level as SOM, which does not need time-consuming training processes. Three performance metrics are calculated as follows:Precision: the proportion of the FECG segments correctly predicted as one class in all the FECG segments predicted as the class.Recall: the proportion of the FECG segments correctly predicted as one class in all the FECG segments labeled as the class.F1-score: the harmonic mean of precision and recall.

Since the methods we use are all unsupervised clustering methods, it is acceptable for the training set to be exactly the same as the test set. A balanced training set is used for classification including 271 low-quality segments, 271 medium-quality segments, and 271 high-quality segments.

#### 4.5.2. For Improvement of FHR Estimation

To assess the performance of improvement of FHR estimation, we compare the results with and without the removal of detected low-quality signal segments. As for the method for FHR estimation, we use the algorithm proposed by the winner of the 2013 Physionet/Computing in Cardiology Challenge [40]. The original input of this method is 4-channel ECG, and we enable the input to be adjusted to one channel FECG so that we can evaluate the performance of improvement of FHR estimation using each channel of our recordings. Furthermore, the usability of every channel is evaluated by the indicator proposed in [40] for selecting the best channel. As for this indicator, the smaller value means higher usability. Thus, based on the evaluation results shown in Figure 5, only recordings extracted from the leads 1, 2, 3, 7, 8, 9, and 11 are utilized for evaluation. We calculated three performance metrics given as follows:Root mean square error (RMSE) between the estimated fetal RR interval (FRRI) value $\hat{FRRI}$ and the reference value FRRI, which is given as follows:
(4)$$RMSE=\sqrt{\frac{1}{\Gamma }\sum_{i=1}^{\Gamma }\left|\right|{\hat{FRRI}}_{i}-FRRI_{i}{\left|\right|}_{2}^{2}},\;$$
where Γ is the number of the FECG signal segments for the estimated FRRIs.Averaged absolute error (AAE) between the estimated value $\hat{FHR}$ and the reference value FHR
(5)$$\begin{array}{cc} AAE & =\frac{1}{\Gamma }\sum_{i=1}^{\Gamma }\left|\right|{\hat{FHR}}_{i}-FHR_{i}\left|\right|\\ & =\frac{1}{\Gamma }\sum_{i=1}^{\Gamma }\left|\right|\frac{60\cdot F_{s}}{{\hat{FRRI}}_{i}}-\frac{60\cdot F_{s}}{FRRI_{i}}\left|\right|, \end{array}\;$$
where $F_{s}$ denotes the sampling frequency of the FECG recordings.The removal rate, which is the proportion of the removed FHRs through the method over all the estimated FHRs.

## 5. Results and Discussion

### 5.1. Feature Evaluation

Figure 6 illustrates the MCFS scores of twelve features. As shown in Figure 6, the novel feature proposed by us, called AE_MSE, achieved the second highest MCFS score of all the extracted features, which indicates that our proposed novel feature plays a significant role in the FECG SQA task. Relatively, five features, including SampEn, HFD, kSQI, sSQI, and AE_MSE, have higher MCFS scores than the other features. Thus, these features may have a greater correlation with FECG signal quality and they may be effective for the FECG SQA task. For the two categories of features, entropy-based and statistical features, only one feature is selected for each category based on the MCFS method. For the features in the same category, it is possible that they are correlated. It is a strategy to reduce redundancy among the correlated features using feature selection. As for the selected five features, they do not have too much relevance and can contribute to the FECG quality level classification task independently.

### 5.2. Classification Evaluation

Table 1 demonstrates the classification results using different unsupervised clustering methods and Figure 7 shows the confusion matrix of results using these methods. By observation of Table 1 and Figure 7, it is obvious that SOM reaches the best performance, particularly the sensitivity for detecting low-quality data (the recall of low-quality data). Higher sensitivity signifies that more low-quality data can be correctly detected rather than classified as high- or medium-quality. While the sensitivities of other methods are all below 70%, SOM achieves 88%, which is much higher. Meanwhile, the precision and recall of high- and medium-quality still maintain a high level within the acceptable range. It is worth noticing that when using SOM, the precision, recall, and F1-score of high-quality data are much higher than using other methods. In other words, a SOM-based classifier can make sure that few high-quality data are misclassified. In practice, the advantages of the SOM-based classifier mentioned above can reserve high-quality signal segments and detect and remove low-quality signal segments as many as possible, which is what we expect.

### 5.3. Improvement of FHR Estimation

As shown in Table 2, for each subject, we calculate the average RMSE, AAE, and removal rate over seven used channels. It can be indicated that RMSE and AAE are all reduced by removing the detected low-quality segments; in the meantime, the removal rate is controlled within an acceptable level. In our dataset, there are many low-quality signal segments in FECG recordings of subject 1 and subject 2. In Table 2, it can be shown that the RMSE and AAE of subjects 1 and 2 are sharply decreased with relatively higher removal rates. Additionally, as displayed in Figure 8, the estimated FHRs with large errors in recordings of subjects 1 and 2 are exactly removed by our proposed method.

### 5.4. Limitation and Future Works

Despite the fact that our method can identify high-, medium-, and low-quality FECG signal segments and thus improve the accuracy of FHR estimation, there are several limitations remaining:

#### 5.4.1. About the Number of Quality Levels

Only three quality levels are defined in our work; however, more quality levels have been mentioned in previous research [11]. A more detailed classification of quality levels may enable the FECG SQA approach to be applied more widely. In our research, it cannot be implemented for now, which is caused by two reasons: (i) Our professional staff has no detailed rules on how to classify multiple quality levels; (ii) our dataset is too small to be divided into several quality levels. Therefore, the next step of our research may focus on enlarging the FECG dataset and setting a clear rule for annotating more than three quality levels.

#### 5.4.2. About the Unbalanced Dataset

Although we balance our dataset for SOM-classifier training simply in this proposal, the size of the training set is sharply decreased because of the reduction in high- and medium-quality samples. Generating more low-quality data rather than just deleting high- and medium-quality data can enlarge the training set to improve the accuracy of classification. Thus, to solve this problem, we may try to generate some fake low-quality data based on real ones in the future.

#### 5.4.3. About the Coverage

In our proposed method, we simply remove estimated FHRs generated from detected low-quality signal segments, which may result in a reduction in the coverage of FHR estimation results for each recording. The discontinuous low-quality signal segments are generally corrupted by occasional noises. Thus, rather than removing them, it is better to recalculate the FHRs estimated from these segments based on adjacent FHRs estimated from high-quality segments. As a result, it is better to propose an approach for interpolating FHRs estimated from low-quality segments in future work.

#### 5.4.4. About Practical Application

For now, we just perform some offline experiments. In the future, we will practically apply this algorithm to the systems and devices. Since clinicians generally have limited knowledge of computer science, we will make the FECG SQA algorithm a black box. The FECG signal segments detected as high-, medium-, and low-quality data can be seen on the monitor, and clinicians can report if there is any incorrect result to help computer scientists to further improve the accuracy of FECG SQA.

## 6. Conclusions

An unsupervised learning-based FECG SQA method was proposed in this paper. The method extracts twelve features, including an AE-based feature, entropy-based features, statistical features, and ECG SQIs, from 1D processed FECG signals and corresponding 2D spectrograms. Subsequently, some significant features are selected using MCFS. In addition, a SOM-based classifier distinguishes high-quality, medium-quality, and low-quality FECG signal segments using these selected features as input. The experimental results demonstrated that, in the three-level FECG SQA, our proposal achieved a weighted average F1-score of 90%. Furthermore, the errors in FHR estimates were somewhat decreased with the acceptable removal rate of identified low-quality signal segments. In addition, because our proposal is based on unsupervised learning, it is totally possible that our method can be used to evaluate the quality of other biomedical signal data.

## Figures and Tables

**Figure 1 bioengineering-10-00066-f001:**
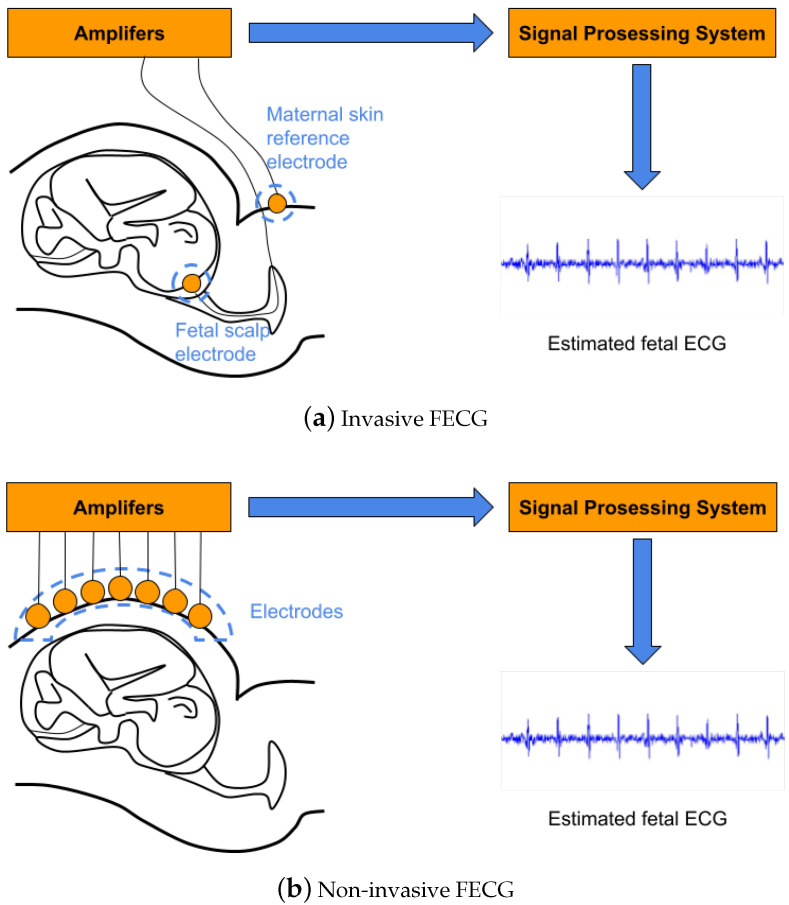
Invasive and non-invasive FECG measurement. (**a**) The invasive FECG signals are extracted by placing the electrode on the scalp of a fetus. (**b**) The non-invasive FECG signals are recorded from the electrodes placed on the maternal abdomen using portable, wearable devices.

**Figure 2 bioengineering-10-00066-f002:**
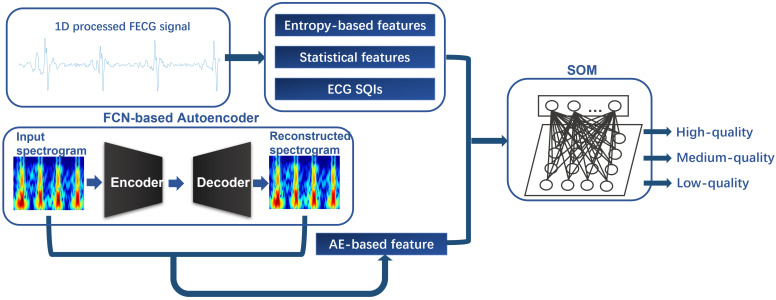
The overall structure of our proposal. The 1D processed FECG signal segments are used to extract entropy-based features, statistical features, and ECG SQIs. Moreover, the 2D spectrograms generated from FECG signals are used to extract an AE-based feature associated with an FCN-based autoencoder. These extracted features are fed into SOM and three quality levels can be identified.

**Figure 3 bioengineering-10-00066-f003:**
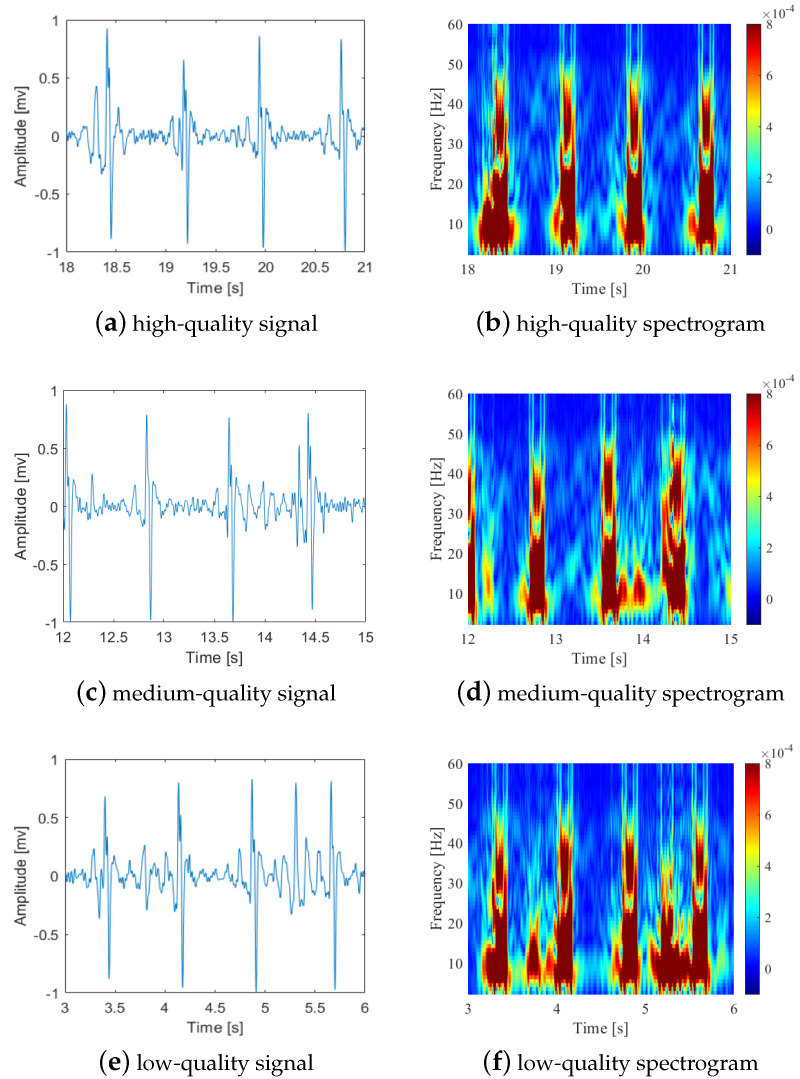
Examples of high, medium, and low-quality FECG signal segments and their corresponding spectrograms. Compared with the high-quality signal segment and spectrogram, the medium-quality one contains some tiny noise, and the low-quality one contains much more noise interference. (**a**,**b**) high-quality FECG signal segments and corresponding spectrogram; (**c**,**d**) medium-quality FECG signal segments and corresponding spectrogram; (**e**,**f**) low-quality FECG signal segments and corresponding spectrogram.

**Figure 4 bioengineering-10-00066-f004:**
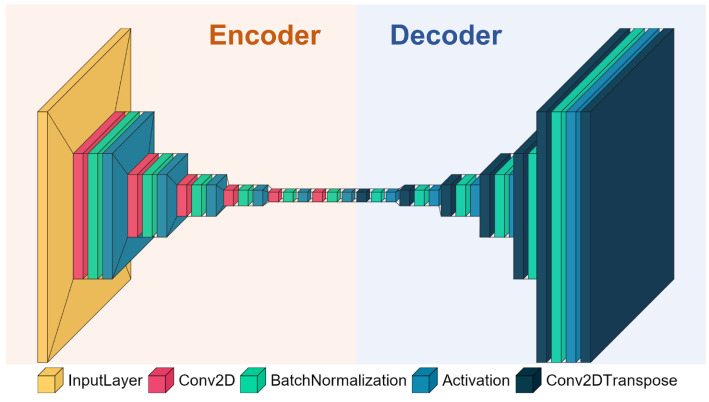
The structure of AE. There are two parts: encoder and decoder. A 2D convolutional (or deconvolution) layer, a batch normalization layer, and an activation layer are composed of an FCN layer. Both in the encoder and the decoder, there are 6 FCN layers.

**Figure 5 bioengineering-10-00066-f005:**
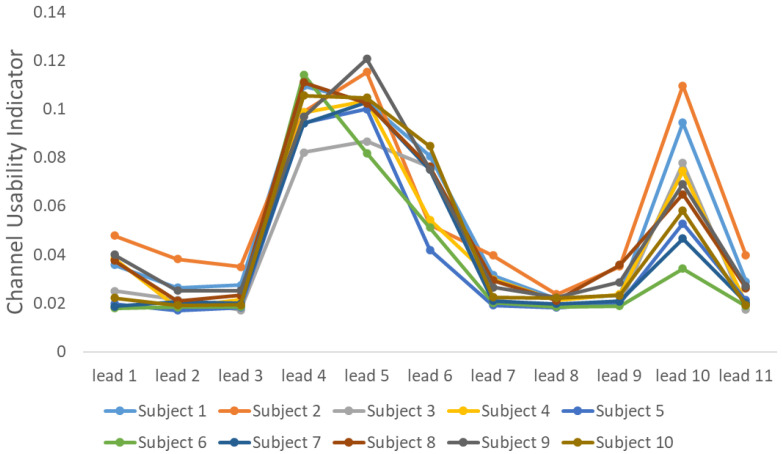
Channel usability evaluation. The smaller value of the channel usability indicator means the higher usability.

**Figure 6 bioengineering-10-00066-f006:**
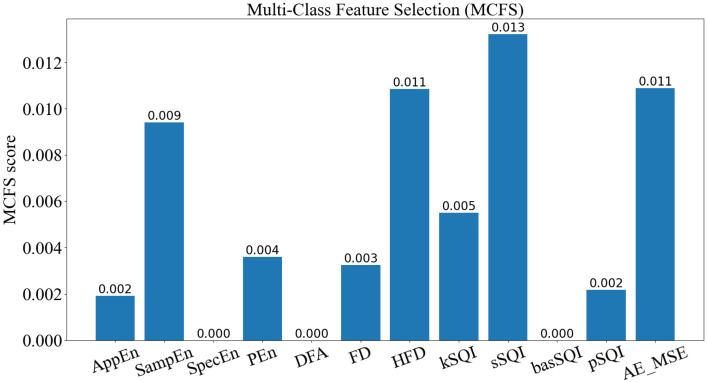
MCFS score for each feature. A feature with a larger score means that this feature plays a more significant role in the classification task. Relatively, five features, including SampEn, HFD, kSQI, sSQI, and AE_MSE, have higher MCFS score than the other features.

**Figure 7 bioengineering-10-00066-f007:**
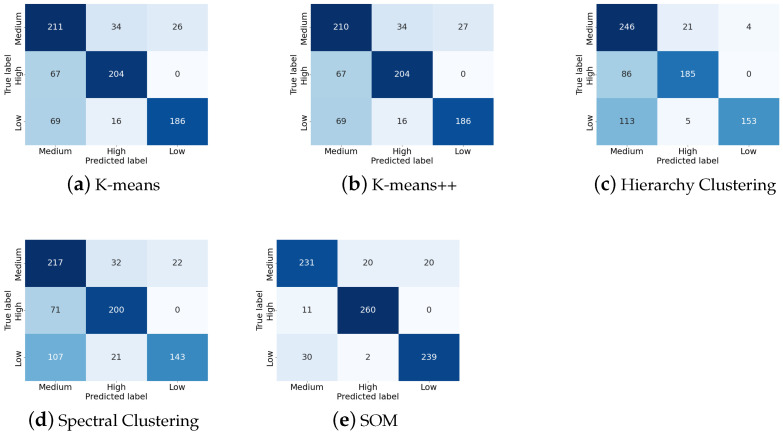
Confusion matrix using five different methods.

**Figure 8 bioengineering-10-00066-f008:**
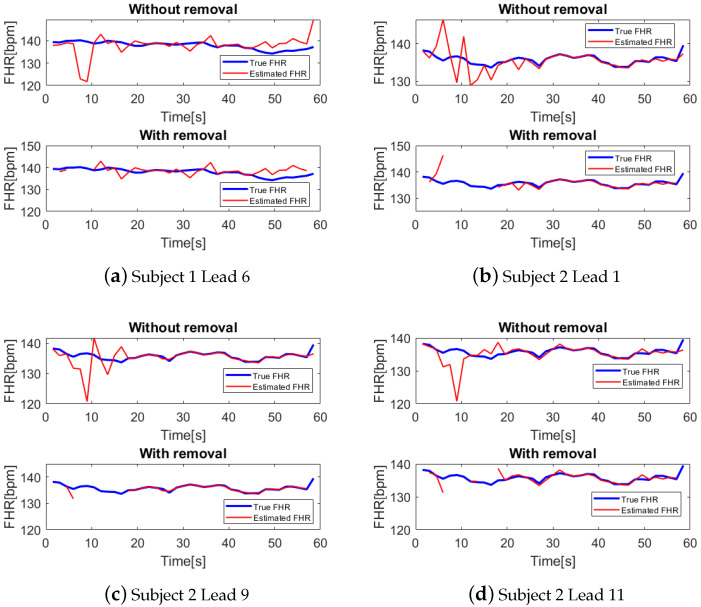
Examples of FHR estimation results with and without removal. It is obvious that the estimated FHRs with large errors are correctly removed by our proposed method.

**Table 1 bioengineering-10-00066-t001:** Summary of the results obtained using different clustering methods. The F1-score of high-, medium-, and low-quality data using SOM are the highest among all five methods. The bold numbers represent the highest value of each performance indicator.

		Precision	Recall	F1-Score
K-means	Medium	0.61	0.78	0.68
	High	0.80	0.75	0.78
	Low	0.88	0.69	0.77
	Average	0.76	0.74	0.74
K-means++	Medium	0.61	0.77	0.68
	High	0.80	0.75	0.78
	Low	0.87	0.69	0.77
	Average	0.76	0.74	0.74
Hierarchy Clustering	Medium	0.55	**0.91**	0.69
	High	0.88	0.68	0.77
	Low	**0.97**	0.56	0.71
	Average	0.80	0.72	0.72
Spectral Clustering	Medium	0.55	0.80	0.65
	High	0.79	0.74	0.76
	Low	0.87	0.53	0.66
	Average	0.74	0.69	0.69
SOM	Medium	**0.85**	0.85	**0.85**
	High	**0.92**	**0.96**	**0.94**
	Low	0.92	**0.88**	**0.90**
	Average	**0.90**	**0.90**	**0.90**

**Table 2 bioengineering-10-00066-t002:** Comparison of RMSE [ms] of FRRI, AAE [bpm] of FHR, and Removal Rate. The RMSE and AAE are all reduced by removing the detected low-quality segments, in the meantime, the removal rate is controlled within 15%.

	No Removal		With Removal		
**Subject**	**RMSE [ms]**	**AAE [bpm]**	**RMSE [ms]**	**AAE [bpm]**	**Removal Rate**
1	0.0047	0.4835	0.0033	0.3045	21.61%
2	0.0075	1.0433	0.0038	0.4722	27.47%
3	0.0018	0.3687	0.0016	0.3472	8.42%
4	0.0021	0.3447	0.0020	0.3334	9.52%
5	0.0033	0.3478	0.0029	0.3212	7.69%
6	0.0010	0.2133	0.0010	0.2043	10.26%
7	0.0019	0.2915	0.0011	0.2264	13.19%
8	0.0191	1.2037	0.0193	1.0410	9.52%
9	0.0078	1.0200	0.0075	0.9871	12.09%
10	0.0036	0.3664	0.0011	0.2130	12.09%
**Average**	**0.0053**	**0.5683**	**0.0044**	**0.4450**	**13.18%**

## Data Availability

The data presented in this study are from Atom Medical Corporation. The data are not publicly available.

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
