# Peer review of "Unsupervised Learning-Based Non-Invasive Fetal ECG Muti-Level Signal Quality Assessment"

_bioengineering, 2023, doi:10.3390/bioengineering10010066_

Round 1

Reviewer 1 Report

1. Authors have discussed Unsupervised learning-based non-invasive fetal ECG multi-level signal quality assessment. But, it has certain observations. Firstly, for Section 1, the authors should provide comments on the cited papers after introducing each relevant work. In addition to it, a general introduction to classification supervised, and unsupervised learning is to be discussed. 

2. In addition, the authors also should provide a more sufficient critical literature review to indicate the drawbacks of existing approaches, then, well define the mainstream research. 

3. Authors must discuss how did the previous studies perform and their limitations? The below papers have some interesting implications that you could discuss in your introduction and how it relates to your work.

doi.org/10.1007/s11042-022-13776-1

4. The writing of the manuscript. There are many incomplete sentences or sentences without subjects. Many sections heading like preliminaries start with a lowercase letter.

5. Authors must state their contributions clearly. In addition to it, what are the limitations of the existing work must be clearly stated.

6. The future scope of the existing work must be stated. 

Reviewer 2 Report

THis paper presents an interesting study on a relevant topic. The manuscript generall reads well and the M&M and the approach adopted for evaluatuion are presented along with the results. I found the authors observations on the limitations in their study both interesting and useful when considering the reported study.

I have comments:

1) The paper has a desriptive title and appropriate keywords (index terms). However, while the use of the acronym in the title is acceptable (it is well known and recognised in the field) the use of acronyms in the abstract is not acceptable and they should be removed with acronyms restricted to the main body text with definitions on first use.

2) The captions to the figures and tables must be improved with suitable explanatory discussion on the figures and tables.

3) It would be useful if the numbered equations were refered to in the related text.

4) The current Introduction is not acceptable as it is in reality bith an introduction and a (too brief) literature review. The introduction must be revised and rewritten in two dedicated sections: (a) Introduction setting out the background and motivation for the study along with a brief overview of the proposed method, the climed contributuin, and a paper structure, and (b) a review of related research with an analysis (the references, while Aadequate may be improved).

5) There are reports that argue that only around 1/3 of published studies are repeatable by other research groups. Accordingly, to provide a basis upon which the reproducibility of the proposed methid may be assessed the authors must make available to research groups the full dataset.

6) The authors must address the topic of practical managerial significance given the paper's medical focus given the collaborative nature of such research and the fact that clinicians generally have limited computational knowledge and computer scientists have limited medical knowledge. 

The manuscript is generally logically structured with a clear narrative (the current introduction needs complete revision) and when suitably revised will in my view be of interest to the intended audience and it will be a suitable candidate for publication.

Reviewer 3 Report

This manuscript investigates a machine learning based signal quality assessment method for fetal ECG. It is of interest and merit for the fetal ECG research community. However, the following concerns should be addressed before accepted for publication.

1.   Lines 167-168. The signals are annotated by medical experts and divided into 3 categories: low-, medium-, and high-quality. However, the authors do not report the criterion for such classification. To make the authors’ method repeatable, the criterion should be described.

2.   Line 163. The authors are suggested to describe the device name rather than the company name ‘Atom Medical Corporation.’ The company name and its address can be noted after the device name.

3.   The quality of fetal ECG is very important. If the quality of the extracted fetal ECG is undesired, then the quality assessment of fetal ECG may be of less meaning. Please describe the technique that the Atom Medical Corporation’s device uses for fetal ECG extraction, and ensure the quality of the extracted fetal ECG.

4.   Line 10. ‘high and low-quality’: should be changed to ‘low- and high-quality.’ Please correct other similar occurrences.

5.   Line 35: ‘by G. D. Clifford et al.’: should be changed to ‘by Clifford et al.’ Please correct other similar occurrences.

6.   Line 115. ‘SFTF’ should be corrected to ‘STFT.’

7.   Line 190. ‘which is mostly can be acceptable’: there is an error in this sentence. Please proofread the whole manuscript.

8.   Figure 3. Color bar is suggested to show for the spectrogram, to make the readers understand different colors correspond to different amplitudes.

9.   Line 272. ‘In the other words’ to ‘In other words.’

10. Line 350. ‘it can not’ to ‘it cannot.’

Round 2

Reviewer 1 Report

1. First of all authors have not fully revised their paper as per the suggestions given in the first review.

2. In section 3.3, the authors used feature extraction and presented some graphs. But, what features are present in the data is not given. In addition to that, how many features are selected is not addressed. Besides, If the method selects k features, what are those features and whether they are relevant or not, must be discussed. 

3. Additionally, what is entropy-based feature and how it is different from statistical features is not addressed. What are statistical-based features, why it is relevant to the study, and what way they benefit must be addressed?

4. Authors have used  K-means, K-means++, hierarchy clustering, spectral clustering, and SOM for classification. But, why the basic classification techniqueques are considered for analysis? It is to be noted that there are several advanced clustering and classifications available in the literature. This is to be ddressed. 

5. In my view, a through revised work is highly required.

Reviewer 2 Report

I have read the author response and the revised manuscript. In general I am content with the revisions; however, there remain two areas of concern:

1) The Introduction has not been revised satisfactorily and it remains a concern. I commented that there must be two sections: (1) the Introduction, and (2) a section addressing an improved literature review. This comment has not been adequately addressed and the original comment remains unresolved.

2) I note the reply to my comment relating to the dataset. I appreciate the nature of the study but, as with other practically implemented studies (much better than studies only based on simulations) I see no reason why the data in a suitably anonymised form (a general principle) cannot be made available.

The outstanding revisions are relatively minor but are required.

Reviewer 3 Report

Thanks for the revision. My concerns have been addressed.
